# The association of cobalturia with cobaltism symptoms a prospective blinded study of 229 post-arthroplasty patients

Stephen S. Tower[1]*, Bradford D. Gessner[2], Christina S. Cho[1], Robert L. Bridges[3]

1 University of Alaska, Medical School, Anchorage, Alaska, United States of America, 2 EpiVac Consulting Services, Anchorage, Alaska, United States of America, 3 Aegis Imaging Consultants, Girdwood, Alaska, United States of America

* sstowertop@gmail.com

**Data Availability Statement:** All relevant data are within the manuscript and its Supporting information files attached to this submission.

## Abstract

### Introduction

Cobalt is a mitochondrial toxin, clinical cobaltism manifests with constitutional, neurologic, and cardiovascular symptomatology. Cobalt's severe toxidrome is known through case reports from extreme wear or corrosion of cobalt-chromium arthroplasty components. However, the spectrum and epidemiology of orthopedic-implant cobaltism and its relationship to duration and degree of cobalt exposure are not well defined.

### Methods

The relationship of urine-cobalt concentration and duration of exposure to cobalt-chromium joint implants and cobaltism symptomatology were prospectively studied in 229 patients. Subjects received a Cobaltism-Symptom-Inventory-Score (CSIS) based on a protocolized interview and examination followed by a spot urine-cobalt measurement.

### Results

129 (56%) subjects were cobalturic (urine-cobalt ≥1.0 ppb). 122 (53%) subjects had a CSIS of >2, this status significantly associates with cobalturia. Median [IQR] urine-cobalt in the subjects with a CSIS >2 was 4.1[1.1–17.0] ppb compared to 0.5[0.5–1.4] ppb in subjects with CSIS ≤ 2. Cobalturia has a sensitivity of 0.69, a specificity of 0.77, and a positive predictive value of 0.74 for a CSIS of >2. The product of years-exposed to a cobalt-chromium implant and urine-cobalt by quartiles significantly positively associates with the Cobaltism-Symptom-Inventory-Score.

### Conclusion

A urine-cobalt of ≥1 ppb likely indicates adverse systemic exposure to orthopedic-implant generated cobalt. Cobaltism severity as quantified by the CSIS significantly correlates with the product of spot urine-cobalt concentration and years-exposed to a cobalt-chromium orthopedic-implant indicating a dose-response relationship. Medical provider and public

**Funding:** The authors received o specific funding for this work.

**Competing interests:** The authors have declared that no competing interests exist.

awareness of orthopedic-implant cobaltism is vital because tens-of-millions are at-risk and early cobaltism is reversible. Further use of cobalt-chromium orthopedic-implants should be questioned given cobaltism becomes clinically apparent at a spot urine-cobalt of 1 ppb or greater. Monitoring of patients with high-risk cobalt-chromium orthopedic-implants appears to be indicated.

## Introduction

The toxidrome of cobalt (cobaltism) primarily manifests as encephalopathy, neuropathy, cardiomyopathy, and endocrinopathy [1–3]. Cobalt exposure from wear or corrosion of orthopedic-implants is the most common etiology of cobaltism with an at-risk population of tens-of-millions [2,4,5]. Cobalt is a mitochondrial toxin that afflicts metabolically active organs, most significantly the brain and the heart [1,6–8]. Cobaltism was first noted sixty-years ago from hip replacement [9], vocational exposure [1,10], cobalt-chloride hematemics [1,11], or cobalt laced beer [1,12]. Cobaltism from modern primary hip replacement was first reported in Alaskans in 2010 [13,14], then Australians in 2011 [15].

Given the large population at-risk and cobaltism's protean manifestations primary medical providers, neurologists, cardiologists, sleep and ENT specialists, psychiatrists, and toxicologists need to be aware that cobaltism might be their patients' unifying diagnosis. Orthopedic publications and organizations consider orthopedic-implant cobaltism to be a rare consequence of extraordinary circumstances [16]. Therefore, of the multiple medical providers that a cobalt-toxic patient might be consulting the implanting surgeon may be the least likely to make the correct diagnosis. Periprosthetic complications of cobalt-chromium metallosis, Adverse Reactions to Metallic Debris (ARMD), has greater awareness among surgeons than its systemic consequence (cobaltism) [17–19]. Several cobalt-chromium hip components at particularly high-risk to generate cobalt-chromium metallosis have been removed from the market due to a high prevalence of ARMD related revision surgery [20,21]. Monitoring programs utilizing cross-sectional imaging (MRI, CT, or ultrasound) and blood-cobalt levels to set criteria for revision surgery were established for these recalled devices [20,21]. Series of patients requiring revision of these implants delineate the periprosthetic complications of cobalt-chromium metallosis without screening for coincident cobaltism [20,21]. Most cases of severe orthopedic-implant cobaltism present in neurologic or cardiovascular extremis of unknown cause [2,3]. Subsequent workup notes blood-cobalt elevations and symptoms or imaging anomalies at the malfunctioning implant leads to removal of worn or corroded cobalt-chromium components resulting in declining blood-cobalt levels with improved systemic symptomatology [2,3].

Endogenously liberated cobalt from wear or corrosion of orthopedic-implants is largely renally excreted but fractionally accumulates in the brain, nerves, heart, liver, kidneys, and cerebrospinal fluid [2,3,6,9,13,22–24]. Overt cobaltism is a catastrophic illness manifesting as Parkinsonism [25], blindness and deafness [22,23], polyneuropathy [24], cognitive and constitutional decline [26], heart transplantation [27], metabolic dysfunction [8], and death [2,3,28]. Unfortunately, because of unawareness, cobaltism may elude diagnosis until profound irreversible deafness and blindness, heart failure requiring transplant, or death occurs [2,3,27]. Early cobaltism symptoms are non-specific and easily misattributed to aging, primary psychopathy, idiopathic cardiomyopathy, primary sleep disorder, or dementia [2–4,6,13,29–33]. Early cobaltism is reversible [2,4,6,13,15,29,30,32,33], rendering diagnosis at this stage essential. Because awareness of cobalt's toxidrome is largely known from extreme case-reports

patients experiencing a lesser illness are likely to be misdiagnosed and therefore not optimally monitored or treated [2,3,25,27].

The relationship between symptomatic cobaltism, urine-cobalt, blood-cobalt, and years-exposed to periprosthetically generated, systemically circulated cobalt lacks prospective investigation. We therefore screened patients with hip, knee, or shoulder cobalt-chromium components for cobaltism symptoms with a novel instrument utilizing a focused history and physical then a spot urine-cobalt.

## Method

229 consecutive patients post-implantation of hip, knee, or shoulder cobalt-chromium components were screened between April 2015 and June 2019. Cohort subsets are reported in other publications [4,29,30]. All subjects provided written consent but because we analysed redacted data University of Alaska IRB designated this inquiry as "Not Human Subject Research" on May 2nd 2019. The study database was redacted per IRB instructions in January of 2020 and was last accessed June 15th 2023. No subjects were excluded from study.

### The Cobaltism-Symptom-Inventory (Table 1)

We adapted an industrial cobaltism questionnaire [34], and an inventory used for a systematic arthroprosthetic cobaltism literature review [2], into a provider administered and scored instrument. Our survey is novel and is not subject of prior study. We relate the Cobaltism-Symptom-Inventory-Score (CSIS) of >2 to cobalturia (urine-cobalt ≥1 ppb) and the CSIS to the product of urine-cobalt concentration and years-exposed to a cobalt-chromium orthopedic-implant.

### Subjects, venue, interview and examination

The surgeon-author evaluated his patients at a community orthopedic practice focused on hip, knee, and shoulder arthroplasty. Patients were included if they had any indwelling hip, knee, or shoulder cobalt-chromium components, no patients were excluded. History included the ten domains of the Cobaltism-Symptom-Inventory (CSI, Table 1). Physical exam included tremor assessment with the subject seated, elbows at side flexed 90 degrees, forearms supinated, and fingers extended. The CSIS was calculated at the end of the encounter then a spot urine-cobalt was obtained. Eight domains (Fatigue, Forgetfulness, Disordered Mood, Disordered Sleep, Imbalance or Weakness, Audiovestibular Dysfunction, Non-refractive Visual Dysfunction, and Generalized Pain) are scored as 0 or 1 (within or beyond the patient's expectations for aging). Tremor/ Neuromotor is scored 0 if tremor was not noted by examiner or patient, 1 if tremor is noted on examination but not historically, 2 if the subject reports daily

**Table 1. The Cobaltism-Symptom-Inventory (CSI).**

| Fatigue | Forgetfulness | Disordered Mood | Disordered Sleep | Imbalance or Weakness | Numbness Extremities | Deafness or Tinnitus | Tremor/ Neuromotor | Non-refractive Visual Dysfunction | Generalized Pain |
|---|---|---|---|---|---|---|---|---|---|
| 0 Expected 1 Excessive | 0 Expected 1 Excessive | 0 Expected 1 Excessive | 0 Expected 1 Excessive | 0 Expected 1 Excessive | 0 None 1 Minor 2 Moderate 3 Severe | 0 Expected 1 Excessive | 0 None 1 Minor 2 Moderate 3 Severe | 0 Expected 1 Excessive | 0 Expected 1 Excessive |

Non-zero score only recorded if patient noted symptom onset or worsening after implantation of a cobalt-chromium arthroplasty component and if patient considered the symptom beyond his/her expectation for aging.

tremor, and 3 for functional impairment from tremor or a neuromotor diagnosis. Numbness is scored 0 for no symptoms, 1 for weekly hand or foot dysesthesia, 2 for daily dysesthesia, and 3 for diagnosed peripheral neuropathy. The CSIS potentially ranges from 0 to 14. A domain is scored non-zero only if its onset occurred or it worsened beyond the patient's aging expectations post-implantation of a cobalt-chromium hip, knee, or shoulder arthroplasty component.

### Spot urine-cobalt determination

Spot urine-cobalt levels were obtained after CSI scoring. Therefore, the interview and CSI scoring were blinded to urine-cobalt result. Urine-cobalt concentrations were determined by Inductively-Coupled-Mass-Spectrometry by commercial laboratories [4,35]. Over the duration of study the reporting threshold for urine-cobalt concentration ranged from 0.2 to 1.0 parts-per-billion (PPB). Patients abstained from vitamin B-12 supplements for two weeks before cobalt testing because high doses of cyanocobalamin elevate urine-cobalt [1,36]. Subjects with a subthreshold urine-cobalt result were assigned a concentration of half of the respective reporting-threshold for the purpose of determining the relationship of the CSIS to the product of urine-cobalt concentration and years-exposed to a cobalt-chromium implant. We choose a value of $\geq 1$ ppb to define cobalturia because this threshold represents the 95$^{th}$ percentile in subjects without cobalt-chromium arthroplasty implants and this degree of cobalturia is associated with objective measures of encephalopathy [30,37], retinopathy [38], cardiomyopathy [39], and audio-vestibular dysfunction [40]. This degree of cobalt exposure is also associated with an increased incidence of clinical cardiomyopathy [41], metabolic syndrome [42], and Obstructive Sleep Apnea (OSA) [33].

### Other data

Demographics, CSIS, urine-cobalt concentrations, years-exposed to a cobalt-chromium implant, arthroplasty types and locations, and implanting surgeon were entered into a spreadsheet (Microsoft Excel 14.7.2 for Mac OS).

### Statistical analysis

Two-tailed Fischer's exact test was used for contingency table analysis. Two-tailed Students T-test was utilized for parametric data and Mann-Whitney test for nonparametric data. A p-value of $\leq 0.05$ is considered significant. Statistical analysis was performed with Prism 9.3.1 for Mac OS.

## Results

The median [IQR] years-exposed to a cobalt-chromium implant was 10 [5–15], age at time of first arthroplasty with a cobalt-chromium component 59 [53–65] years, patient weight 86 [73–102] kilograms, and 47% of the subjects were female (Table 2).

The surgeon-author or a local colleague performed 179 (78%) of the at-risk arthroplasties. Table 3 lists the location, type, location, brand and model of the various cobalt-chromium components and the risk-level of their implantation to result in cobalturia as categorized in another publication studying the same patient cohort [4].

Only 4 (2%) of the 229 subjects had a formally recalled arthroprosthetic device (Stryker Rejuvenate femoral stem with modular cobalt-chromium neck) [43]. 128 subjects (56%) were cobalturic (urine-cobalt of $\geq 1$ pbb). Urine-cobalt levels were above reporting-threshold for 179 (78%) subjects, the 50 (22%) of subjects with a sub-threshold result were assigned a urine-cobalt of half of the reporting-threshold. The mean (range) and median [IQR] urine-cobalt of

**Table 2. Demographics of cobalturic and not-cobalturic groups.**

| Group | N (Percent) | Age at time of arthroplasty Mean (range) Median [IQR] | Weight (KG) Mean (range) Median [IQR] | Percent Female | Years-Exposed Mean (range) Median [IQR] | Spot Urine-cobalt ppb Mean (range) Median [IQR] | CSI Score Mean (range) Median [IQR] |
|---|---|---|---|---|---|---|---|
| Cobalturic | 128 (56%) | 58 (25–82) 59 [53–65] | 88 (46–166) 86 [73–102] | 51 | 11 (2–33) 9 [6–15] | 24.0 (1.0–446) 7.0 [2.0–19.0] | 5.0 (0–12) 5 [2–8] |
| Not-Cobalturic | 101 (44%) | 60 (40–83) 59 [53–65] | 87 (40–161)) 84 [74–101] | 44 | 12 (2–32) 11[7–16] | 0.5 (0.1–0.9) 0.5 [0.3–0.5] | 1.8 (0–10) 1 [0–3] |
| Total | 229 (100%) | 59 (25–83) 59 [53–65] | 88 (40–166) 86 [73–103] | 47 | 12 (2–33) 10 [7–15] | 13.0 (0.1–446) 1.3 [0.5–8.6] | 4.0 (0–12) 3 [0–6] |
| Cobalturic compared to Not-Cobalturic | | p = 0.16 | p = 0.53 | p = 0.29 | p = 0.38 | | p< 0.0001 |

all 229 subjects was 13.2 (0.1–446) and 1.3 [0.5–8.9]. The mean CSIS of the cobalturic group of 5.0 is significantly higher (p < 0.0001) than the mean CSIS of 1.8 in the not-cobalturic group, there were no significant differences in age, weight, gender, or years-exposed between the cobalturic and not-cobalturic groups (Table 2, Fig 1).

Cobalturia is sensitive and specific for CSIS of >2 (p <0.0001). A urine-cobalt of ≥1 ppb has a sensitivity of 0.69, a specificity of 0.77, positive-predictive-value of 0.72, negative-predictive-value of 0.74, and a likelihood ratio of 2.8 for a CSIS of >2 (Table 4) comparative to the not-cobalturic cohort.

Cobalturic subjects scored significantly higher in all domains of the Cobaltism-Symptom-Inventory than not-cobalturic subjects (Table 5).

There is a significant increase (p <0.0001) in CSIS between the first and second quartiles of subjects sorted by the product of urine-cobalt concentration and years-exposed to a cobalt-chromium implant and the third quartile, and between the third and fourth quartiles (Table 6, Fig 2).

## Discussion

We find a significant correlation of the Cobalt-Symptom-Inventory-Score with the product of years-exposed to a cobalt-chromium orthopedic-implant and urine-cobalt concentration indicating a dose-response relationship of orthopedic-implant cobalt exposure over time and cobaltism severity as reflected by the CSIS. A urine-cobalt of ≥1 ppb alone is specific, sensitive, and positively predictive for symptomatic cobaltism as indicated by a CSIS of >2. These findings are consistent with other studies with similar degrees of cobalt exposure from cobalt-chromium orthopedic-implants finding quantitative brain hypometabolism [30], brain atrophy [37], retinopathy [38], Obstructive-Sleep-Apnea [33], psychopathy with elevated genotoxic markers [31], audiovestibular dysfunction [40], and echocardiographic cardiomyopathy [39], comparative to matched controls without cobalt-chromium orthopedic-implants. Bridges published on a subgroup of 57 cobalturic subjects of this cohort, all had brain hypometabolism by quantitative FDG-PET assessment compared to age and gender matched atlas controls [30]. Eight-subjects with repeat quantitative FDG-PET brain imaging one-year after revision surgery and/or oral N-acetylcysteine [44–47], normalized cobalt levels exhibited improved brain metabolism and diminished symptomatology [29,30].

Analysis of the American National Health and Nutrition Examination Survey (NHANES) database of 3442 adults found subjects with a blood-cobalt of >0.17 ppb (equivalent to urine-cobalt of 0.7 ppb) [4], have increased prevalence of cardiovascular disease compared to those with a blood-cobalt ≤0.11 ppb [41]. Another NHANES study of 947 women found a

**Table 3. Location and types of cobalt-chromium components.**

| Location | Risk | Type | Brand and Model of cobalt-chromium component | Number Percent of Cohort | Percent Cobalturic | Mean (range) Median [IQR] Urine Cobalt |
|---|---|---|---|---|---|---|
| Hip | Extreme | Metal-on-Metal articulation of Total-Hip-Replacement or Hip-Resurfacing | Zimmer<br>Wright Medical<br>Smith & Nephew<br>Biomet<br>J&J (DePuy) | 37<br>16% | 97% | 54 (0.8–446)<br>9.7 [3.7–28.5] |
| Hip | High | Metal-on-Plastic or Ceramic-on-Plastic hips prone to taper corrosion of junctions involving a cobalt-chromium component | Stryker<br>(V40 taper head)<br>(Modular Dual Mobility Socket)<br>(Rejuvenate Modular Neck)<br>Zimmer<br>(Versys 12/14 taper head)<br>Wright Medical or Microport<br>(Modular Neck) | 136<br>59% | 56% | 6.6 (0.1–55)<br>1.2 [0.5–8.0] |
| Hip | Low | Solitary Metal-on-Plastic hips not prone to taper corrosion involving a cobalt-chromium component | Zimmer Metasul 12/14 head 38 mm<br>Styker of Osteonics C taper head<br>DePuy taper heads<br>Biomet taper heads | 17<br>7% | 0% | 0.4 (0.1–0.8)<br>0.4 [0.3–0.6] |
| Shoulder | High | Hemiarthroplasty<br>Anatomic replacement<br>Reverse replacement | Zimmer<br>J&J (DePuy)<br>Biomet | 11<br>5% | 73% | 6.0 (0.1–24)<br>4.3 [0.5–11] |
| Knee | High | Bilateral knee replacements or revision knee replacement(s) | Zimmer, Stryker, DePuy, Biomet | 11<br>5% | 45% | 3.1 (0.2–18)<br>0.8 [0.4–3.0] |
| Knee | Low | Solitary Primary replacement | Zimmer, Stryker, DePuy | 10<br>4% | 0% | 0.4 (0.1–0.8)<br>0.5 [0.2–0.5] |
| Multiple | High | Multiple low risk arthroplasties<br>Low risk arthroplasty and spinal instrumentation | Many combinations | 7<br>3% | 71% | 1.6 (0.5–3.2)<br>0.8 [0.7–2.5] |
| | | | Combined Cohort | 229<br>100% | 56% | 13.2 (0.1–446)<br>1.3 [0.5–8.9] |

significantly higher prevalence of metabolic-syndrome in subjects with urine-cobalt of ≥0.95 ppb compared to those with urine-cobalt ≤0.38 ppb [42]. Notably, the NHANES's reporting-threshold for cobaltemia and cobalturia is 0.1 ppb and their cohort includes subjects with orthopedic-implants [41,42]. A urine-cobalt of 1 ppb is equivalent to a blood-cobalt of 0.3 ppb [4], which is below the common reporting threshold of 0.5 ppb of commercial laboratories [4]. The degree of systemic cobalt exposure associated with increased prevalence of metabolic syndrome or cardiomyopathy is similar to our threshold of a urine-cobalt ≥1 ppb for orthopedic-implant cobaltism as quantified by the Cobaltism-Symptom-Inventory [41,42].

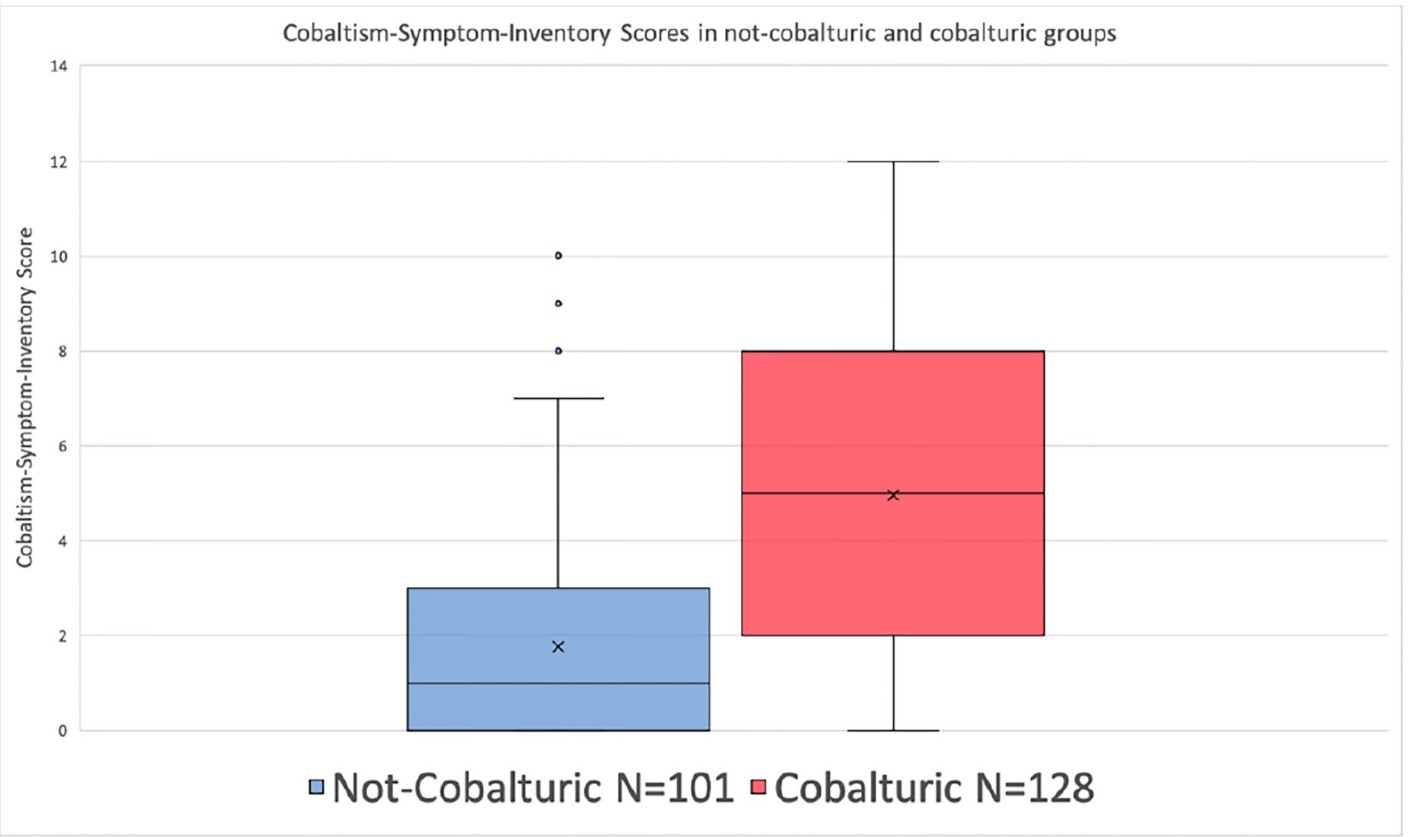

**Fig 1. Cobaltism-Symptom-Inventory Scores (CSIS) in Not-Cobalturic and Cobalturic groups.**

The domains of the Cobaltism-Symptom-Inventory are consistent with the presentation of overt orthopedic-implant cobaltism as described in case-reports and series [2–4,29,30]. Moreover, the components of the CSI reflect symptomatology expected from pathophysiology noted in subjects with systemic cobalt exposure from cobalt-chromium orthopedic-implants [2–4,9,29–31,33,34,37,38,40]. Post-arthroplasty patients with blood-cobalt concentrations >100 ppb may die or experience heart transplantation, profound deafness and blindness,

**Table 4. Contingency analysis of CSIS >2 and urine-cobalt ≥1 ppb.**

|  | CSIS ≤2 N (%) | CSI >2 N (%) | Total |
|---|---|---|---|
| Not-Cobalturic N (%) | 73 (72%) | 28 (28%) | 101 (44%) |
| Cobalturic N (%) | 33 (26%) | 95 (74%) | 128 (56%) |
| Total N (%)<br>Urine-cobalt Mean (Range)<br>Urine-cobalt Median [IQR] | 106 (47%)<br>6.0 (0.1–333)<br>0.5 [0.5–1.4] | 123 (53%)<br>19.4 (0.1–446)<br>4.1 [1.1–17.0] | 229 (100%)<br>13.1 (0.1–446)<br>1.3 [0.5–8.6] |

Fisher's exact Two-sided, P <0.0001, Sensitivity 0.69, Specificity 0.77, Likelihood ration of 2.8, Positive Predictive Value 0.72, Negative Predictive Value 0.74 for subjects with urine-cobalt ≥1 ppb to have Cobaltism-Symptom-Inventory-Score >2.

Mann Whitney test of urine-cobalt concentration of subjects with CSIS ≤2 versus those with CSIS >2 (p <0.0001).

**Table 5. Cobaltism-Symptom-Inventory domain scores in cobalturic and not-cobalturic groups.**

| | Imbalance or Weakness | Forgetfulness | Disordered Mood | Generalized Pain | Disordered Sleep | Fatigue | Extremity Numbness (0–3) | Tremor/ Neuromotor (0–3) | Deafness Tinnitus Vertigo | Non-refractive Visual Dysfunction |
|---|---|---|---|---|---|---|---|---|---|---|
| Non-zero score in Not-Cobalturic Subjects | 21% | 22% | 18% | 12% | 6% | 21% | | | 14% | 2% |
| Non-zero score in Cobalturic Subjects | 45% | 58% | 51% | 27% | 36% | 58% | | | 34% | 16% |
| Mean score in Not-Cobalturic Subjects | | | | | | | 0.3 | 0.7 | | |
| Mean score in Cobalturic Subjects | | | | | | | 0.7 | 1.3 | | |
| Likelihood ratio non-zero score in Cobalturic Subjects | 2.2 | 2.6 | 2.8 | 2.3 | 6 | 2.8 | 2.3 | 1.9 | 2.1 | 8 |
| P value | < 0.0001 | < 0.0001 | < 0.0001 | < 0.0001 | < 0.0001 | < 0.0001 | < 0.0001 | < 0.0001 | = 0.0007 | = 0.005 |

disabling peripheral neuropathy with thyropathy, or career-ending Parkinsonism [2,3,25]. Those with double or single-digit blood-cobalt may experience combinations of psychopathy with elevated serum genotoxic markers [31], cognitive decline, sensory impairments, tremor, fatigue, headaches, and disordered sleep [2–4,29,30].

In our cohort a CSIS >2 relates to cobalturia better than symptoms in any one of its ten domains. For example, a CSIS >2 has a PPV of 0.72 and a NPV of 0.74 of cobalturia. Comparatively, audiovestibular dysfunction (deafness, tinnitus, vertigo) is a relatively poor discriminator for cobalturia (PPV 0.86, NPV 0.34) because interval (post-implantation of a cobalt-chromium orthopedic-implant) audiovestibular dysfunction in a geriatric population has many etiologies including post-viral or post-vaccination syndromes [6,40,48,49].

Retinal atrophy is a marker of clinical and pre-clinical dementia and is easily determined by Optical Coherence Tomography (OTC) [50]. Knee replacement patients with a median blood-cobalt of 1.1 ppb exhibit macular and retinal nerve layer thinning comparative to matched

**Table 6. Comparison of subjects with lower half product of urine-cobalt and Years-Exposed (YE) with Cobaltism-Symptom-Inventory-Score (CSIS) with third and fourth quartiles.**

| Quartiles | N | (Urine-Cobalt) * (YE) Mean (range) Median [IQR] | CSIS Mean (range) Median [IQR] |
|---|---|---|---|
| First and Second Quartiles | 114 | 5.1 (0.3–16.7) 3.9 [2.3–7.3] | 1.0 (0–10) 1 [0–3] |
| Third-Quartile | 57 | 38.6 (17.6–87.0) 32.3 [24.5–47.8] | 4.3 (0–10) 4 [2–7] |
| Fourth-Quartile | 58 | 432 (88.3–3678) 219 [135–408] | 6.0 (0–12) 6 [4–8] |

CSIS of first and second quartiles v third quartile p <0.0001.
CSIS of third quartile v forth quartile p <0.0001.

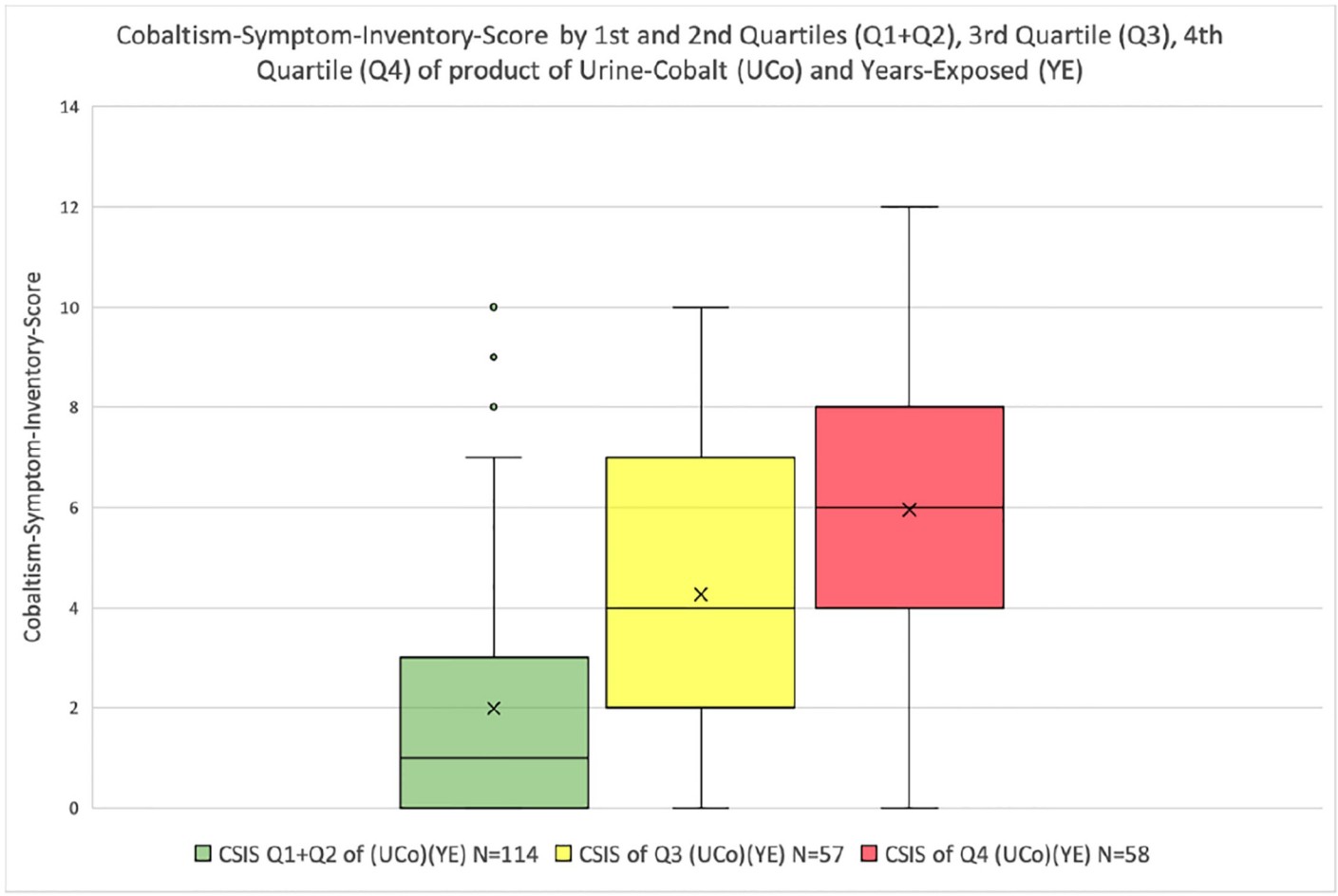

**Fig 2. Plot of Cobaltism-Symptom-Inventory-Score in subjects of first and second quartiles and third and fourth quartiles of the product of urine-cobalt and Years-Exposed to a cobalt-chromium implant.**

controls without cobalt-chromium implants all with blood-cobalt of <0.5 ppb [38]. The frontal-temporal brain hypometabolism found in half of our cobalturic subjects by quantitive FDG-PET may presage dementia [30]. The cellular mechanisms of cobalt encephalopathy are expressed with a level of systemic cobalt exposure consistent with a urine-cobalt concentration of ≥1 ppb [51].

Gessner's 2015 systematic literature review of arthroprosthetic cobaltism identified 25-cases [2], Crutsen's similar 2022 study found 79-cases [3]. Most of the inclusive cases represent the least common and most severe presentations of orthopedic-implant cobaltism [2,3], most subjects had sentinel symptoms or findings at their malfunctioning arthroplasty [2,3]. The tripling of severe case-reports over seven-years likely indicates an increasing awareness of overt orthopedic-implant cobaltism [2,3]. Periprosthetic toxicity of cobalt-chromium metallosis (ARMD) is better known to surgeons of than its systemic consequences (cobaltism) [17–19]. Several devices at particularly high-risk to generate wear or corrosion have been recalled due to a high prevalence of revision surgery for ARMD and for these recalled devices monitoring programs utilizing cross-sectional imaging (MRI, CT, or ultrasound) of the arthroplasty were utilized to set criteria for revision surgery [20,21].

A urine-cobalt of 1 ppb translates to a blood-cobalt of 0.3 ppb which is below the common blood-cobalt reporting threshold of 0.5 ppb of commercial laboratories [4]. For this reason, urine-cobalt is a better screening test for concerning orthopedic-implant cobalt exposure than blood-cobalt [4]. The common commercial laboratory reporting-thresholds for urine-cobalt of 1 ppb and blood-cobalt of 0.5 ppb likely gives clinicians the false impression that these are upper-range "normal" values rather than an outlier concentrations in patients without a cobalt-chromium implant [38,41,42]. Medical providers might misconstrue that a urine-cobalt <1.0 ppb "rules out" cobaltism. Studies of cobalt-levels in subjects before and after cobalt-chromium orthopedic-component implantation with reporting-thresholds of ≥0.1 ppb indicate cobalt-levels generally increase 3 to 10-fold post-operatively dependent upon implant type and design which results in supraphysiologic cobalt exposure yet with urine or blood cobalt concentrations below commercial laboratories' reporting-thresholds [4,38]. Patients implanted with cobalt-chromium orthopedic-implants may experience cobalt exposure with adverse physiologic effect with a urine-cobalt <1 ppb or blood-cobalt <0.5 ppb [41,42].

Only 4 (3%) of our 128 cobalturic subjects had an arthroplasty formally recalled for cobalt-chromium metallosis complications [4]. None of our 37 subjects with extreme-risk metal-on-metal hips had a recalled implant [4]. Therefore, only a fraction of patients fitted with cobalt-chromium orthopedic-implants, or their medical-providers, are likely aware that cobaltism maybe etiologic of their, or their patients', neurologic, constitutional, or cardiac symptoms.

## Strengths and weakness

This is the first blinded prospective survey of at-risk subjects to relate cobaltism symptoms to the degree and duration of orthopedic-implant systemic cobalt exposure. The sensitivity and specificity of a urine-cobalt ≥1.0 ppb for clinical cobaltism indicate this worry-line is appropriate. Although most domains of the CSI involve patient report of common subjective neurologic and constitutional symptomatology the strong correlation of the CSIS to the product of urine-cobalt level and years-exposed to a cobalt-chromium orthopedic-implant indicate that the CSIS likely quantitates clinical cobaltism. The confirmation of the diagnosis of cobalt-encephalopathy in half of our cobalturic patients by quantitative FDG-PET imaging confirms that a urine-cobalt ≥1 ppb relates to neurologic metabolic dysfunction [30].

The generalizability of the Cobaltism-Symptom-Inventory as a screening tool for orthopedic-implant cobaltism bears confirmation in a general medical population. The at-risk implants of our subjects reflect regional preference and some implants result in cobalturia more frequently than others [4], other at-risk populations may not have the same prevalence of cobalturia and resulting cobaltism as our cohort [4]. Subject selection-bias is likely in our study because the surgeon-author is known for his interest in orthopedic-implant cobaltism [2,4,29,30]. However, this is unlikely to be significant confounder given 78% of subjects had their at-risk surgery performed in the surgeon-author's geographically isolated community of 290,000.

## Directions for future research

In order to validate the CSI as a screening tool for orthopedic-implant cobaltism diagnostic criteria need to be established. Optimal criteria would likely include urine-cobalt and/or blood-cobalt levels, the CSIS, and objection signs of end-organ damage such as retinal nerve layer thinning by OTC [38], brain hypometabolism by FDG PET CT [30], brain atrophy by quantitative MRI [37], audiovestibular dysfunction by quantitative audiovestibular testing [40], and heart function by echocardiography [39]. Establishing such criteria would require large prospective studies screening post-arthroplasty patients with the CSI before urine and/or

blood cobalt determination followed by objective assessment of the retina [38], brain structure and function [30,37], audiovestibular testing [40], and echocardiography [39,52,53].

In particular the relationship of the CSIS and echocardiographic measures of systolic and diastolic cardiac function should be explored given that neurologic and cardiac toxicity is noted together in many case reports of arthroprosthetic cobaltism and given the ready availability of echocardiography which does not expose the patient to radiation [2,13,14,39,41,52,53].

Treatment of orthopedic-implant cobaltism is presently empiric and source control is most commonly employed (revision of corroding or worn cobalt-chromium components with ceramic, plastic, or stainless steel or titanium alloy alternatives) for patients with severe toxicity and those with periprosthetic metallosis complications (ARMD) [2,3]. Dietary supplements such as N-acetyl-cysteine, glutathione, and alpha-lipoic-acid that chelate cobalt or ameliorate its metabolic toxicity might be effective for patients with well-functioning arthroplasties (no evidence of periprosthetic complications), mild systemic symptomatology, and minor elevations in blood or urine cobalt-levels [44–47,54]. Determination of optimal treatment strategies will require large prospective studies with quantitation of cobalt levels, cobaltism symptomatology, and end-organ damage before and after treatment.

## Conclusion

The Cobaltism-Symptom-Inventory takes five minutes to administer and score and a spot urine-cobalt level is easily obtained at relatively low cost at commercial laboratories (Quest 68 USD). The large at-risk population with cobalt-chromium orthopedic-implants, the treatability of early cobaltism [4,29,30,44,45,54], and the severity and irreversibly of the severe cobaltism [2,3,27], position orthopedic-implant cobaltism as an ideal malady for screening. Over-the-counter N-acetylcysteine appears to be of benefit to cobalturic patients with mild cobaltism [4,29,30,44,45,54]. Patients with severe cobaltism and metallosis mediated periprosthetic tissue damage benefit from revision surgery and delay in diagnosis and remediative surgery may result in irreversible periprosthetic, neurologic and cardiac pathology [2,3]. Clinician awareness that cognitive decline, constitutional deterioration, neurologic or psychiatric disorders, or cardiomyopathy are associated with cobalturia (urine-cobalt $\geq 1$ ppb) is important because orthopedic-implant cobaltism is likely common and easily treated if diagnosed in its early stages [2–4,15,29–31,38,40,44,45]. Further implantations of cobalt-chromium orthopedic-implants should be carefully considered given safer and equally efficacious alternative materials including cobalt-free alloys, polyethylenes, and ceramics.

Orthopedic-implant cobaltism appears to be a under recognized toxic progeria. Clinician and public awareness of this entity is important to limit associated morbidity given the benefits of prevention (use of cobalt-free orthopedic-implants), screening (spot urine-cobalt), at-risk population recognition (millions of patients implanted with a cobalt-chromium orthopedic-implants), syndrome recognition (use of the Cobaltism-Symptom-Inventory), and a benign treatment option (OTC N-acetyl-cysteine) for patients with nominal cobalt exposure and symptomatology [44–47]. For patients with severe orthopedic-implant cobaltism source control is required to limit neurologic and cardiovascular morbidity and mortality [2,3].

## Supporting information

**S1 Checklist.**
(DOCX)

**S1 Data.**
(XLSX)

**S1 File.**
(PDF)

## Author Contributions

**Conceptualization:** Stephen S. Tower.

**Data curation:** Stephen S. Tower, Bradford D. Gessner.

**Formal analysis:** Stephen S. Tower, Bradford D. Gessner.

**Investigation:** Bradford D. Gessner, Christina S. Cho, Robert L. Bridges.

**Methodology:** Stephen S. Tower, Bradford D. Gessner.

**Supervision:** Stephen S. Tower.

**Writing – original draft:** Stephen S. Tower.

**Writing – review & editing:** Stephen S. Tower, Bradford D. Gessner, Christina S. Cho, Robert L. Bridges.

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
