## [Decision Letter · Decision Letter 0]

14 Sep 2023

PONE-D-23-19885The association of Cobalturia with Cobaltism Symptoms

A prospective blinded study of 229 post-arthroplasty patientsPLOS ONE

Dear Dr. Tower,

Thank you for submitting your manuscript to PLOS ONE. After careful consideration, we feel that it has merit but does not fully meet PLOS ONE’s publication criteria as it currently stands. Therefore, we invite you to submit a revised version of the manuscript that addresses the points raised during the review process.

We look forward to receiving your revised manuscript.

Kind regards,

Antonino Maniaci

Academic Editor

PLOS ONE

2. Please amend your manuscript to include your abstract after the title page.

Additional Editor Comments:

Please perform all revisions required.

Reviewers' comments:

Reviewer's Responses to Questions

**Comments to the Author**

1. Is the manuscript technically sound, and do the data support the conclusions?

Reviewer #1: Yes

Reviewer #2: Yes

2. Has the statistical analysis been performed appropriately and rigorously? 

Reviewer #1: Yes

Reviewer #2: Yes

3. Have the authors made all data underlying the findings in their manuscript fully available?

Reviewer #1: Yes

Reviewer #2: Yes

4. Is the manuscript presented in an intelligible fashion and written in standard English?

Reviewer #1: Yes

Reviewer #2: Yes

5. Review Comments to the Author

Reviewer #1: The paper need major revisions. Perform all the suggestions:

Here are some more detailed suggestions for improving each section:

Introduction:

- Provide 2-3 sentences explaining cobaltism - a clinical syndrome caused by cobalt exposure leading to neurological, cardiovascular, and other manifestations. cite PMID: 21066926.

- line 48, among differential diagnose other causes should be discussed leading to cognitive impairment, fatigue, sleepiness as obstructive apnea. Please cite doi:10.1007/s11325-021-02520-y

- Note that cobalt release from joint implants is a major cause today. Briefly explain mechanisms - corrosion, wear, release into blood/urine.

- State that relationship between symptoms, urine cobalt levels, and exposure time is not well studied.

Methods:

- In developing symptom questionnaire, describe process for choosing domains, consulting existing questionnaires, pilot testing questions.

- For urine cobalt testing, explain choice of 1 ppb cutoff based on prior studies showing associations with clinical effects at this level.

- Describe all statistical tests used - correlations, comparisons between groups, ROC analysis.

Results:

- Report demographics of cohort - age, gender, implant types.

- Present urine cobalt levels summary statistics before analyzing by cobaltism symptoms.

- Use tables/figures to clearly show symptom scores for each domain by cobalturic vs non-cobalturic groups.

Discussion:

- Note limitations including single-center cohort, selection bias, lack of universal cobaltism criteria.

- Compare associations found to prior studies linking urine cobalt to clinical effects.

- line 175, alternative causes of vertigo should be discussed as drugs or viral infections. please Cite doi: 10.1177/0960327111414280 and doi:10.3389/fmed.2021.790931

- Discuss need for multi-center studies to validate questionnaire, urine thresholds.

- Suggest future research on treatment effects tied to urine cobalt reductions.

- Avoid overstating the conclusions - cohort limitations preclude definitive conclusions.

Reviewer #2: I read with interest the manuscript by Tower et al. The paper is original and sound. However, there are some issues that need o be addressed with a revision:

- I could not find the abstract anywhere in the manuscript. Please provide it.

- Line 50-51. Authors should also cite hearing loss and delirium as possible adjunct misdiagnosis of cobaltism (doi: 10.1177/0960327111414280 - doi: 10.3390/jcm12020435). Please briefly discuss and add these 2 references.

- Line 58-64. These sentences summarize both methods and results, and should therefore be not included in the introduction section.

- Please provide ethical committee approval or waiver code and date of approval.

- How did authors retrieve data from patients with dementia? Were they excluded? Please specify.

- Please provide the main investigated outcome in the methods section.

- Lack of data on cardiovascular function of the patients enrolled should be highlighted as a further possible limitation of the study and direction of future research, as echocardiography is nowadays a feasible bedside tool to investigate systolic and diastolic function of hospitalized patients (doi: 10.2106/JBJS.16.00743- doi: 10.1007/s12630-022-02225-0 - doi: 10.1111/echo.15462). Please discuss and add these 3 references.

6. PLOS authors have the option to publish the peer review history of their article (what does this mean?). If published, this will include your full peer review and any attached files.

Reviewer #1: No

Reviewer #2: No

---

## [Author Response · Author response to Decision Letter 0]

27 Sep 2023

PONE-D-23-19885

The association of Cobalturia with Cobaltism Symptoms

A prospective blinded study of 229 post-arthroplasty patients

PLOS ONE

Corresponding Author responds to Reviewers

I appreciate the efforts of the reviewers, edits made to the manuscript will be detailed in red in the context of their critique.

Reviewer #1: The paper needs major revisions. Perform all the suggestions:

Here are some more detailed suggestions for improving each section:

Introduction:

- Provide 2-3 sentences explaining cobaltism - a clinical syndrome caused by cobalt exposure leading to neurological, cardiovascular, and other manifestations. cite PMID: 21066926. See lines 30-38.

- line 48, among differential diagnose other causes should be discussed leading to cognitive impairment, fatigue, sleepiness as obstructive apnea. Please cite doi:10.1007/s11325-021-02520-y. See lines 40-44 and 66-73.

Added references 

26. Asker S, Asker M, Yeltekin AC, Aslan M, Demir H. Serum levels of trace minerals and heavy metals in severe obstructive sleep apnea patients: correlates and clinical implications. Sleep and Breathing. 2015;19:547-52.

- Note that cobalt release from joint implants is a major cause today. See lines 31-38.

Briefly explain mechanisms - corrosion, wear, release into blood/urine. See lines 31-32.

- State that relationship between symptoms, urine cobalt levels, and exposure time is not well studied. See lines 75-79

Methods:

- In developing symptom questionnaire, describe process for choosing domains, consulting existing questionnaires, pilot testing questions. See lines 87-93.

- For urine cobalt testing, explain choice of 1 ppb cutoff based on prior studies showing associations with clinical effects at this level. See lines 114-129.

- Describe all statistical tests used - correlations, comparisons between groups, ROC analysis. See lines 134-137, I believe ROC analysis is not appropriate for this study because there is no gold standard for the diagnosis of orthopedic-implant cobaltism, additionally the quartile analysis for both spot urine [Co] and the product of urine [Co] and years of exposure shows the relationship of both parameters to the CSIS.)

Results:

- Report demographics of cohort - age, gender, implant types. Lines 139-156 and Tables 2&3.

- Present urine cobalt levels summary statistics before analyzing by cobaltism symptoms. Reordered as requested, Lines 158-164 and Tables 2&3.

- Use tables/figures to clearly show symptom scores for each domain by cobalturic vs non-cobalturic groups. See Tables 2, 4, 5, 6, and Figures 1&2.

Discussion:

- Note limitations including single-center cohort, selection bias, lack of universal cobaltism criteria. See Lines 291-337

- Compare associations found to prior studies linking urine cobalt to clinical effects. See Lines 206-216

- line 175, alternative causes of vertigo should be discussed as drugs or viral infections. please Cite doi: 10.1177/0960327111414280 and doi:10.3389/fmed.2021.790931. See Lines 242-247, and added references:

40. Agrup C, Gleeson M, Rudge P. The inner ear and the neurologist. Journal of Neurology, Neurosurgery & Psychiatry. 2007;78(2):114-22.

41. Di Mauro P, La Mantia I, Cocuzza S, Sciancalepore PI, Rasà D, Maniaci A, et al. Acute vertigo after COVID-19 vaccination: case series and literature review. Frontiers in medicine. 2022;8:790931.

- Discuss need for multi-center studies to validate questionnaire, urine thresholds. See lines 291-337.

- Suggest future research on treatment effects tied to urine cobalt reductions. See lines 328-337.

- Avoid overstating the conclusions - cohort limitations preclude definitive conclusions. The adjectives “likely” and “apparent” liberally applied.

Reviewer #2: I read with interest the manuscript by Tower et al. The paper is original and sound. However, there are some issues that need to be addressed with a revision:

- I could not find the abstract anywhere in the manuscript. Please provide it. See lines 1-28.

- Line 50-51. Authors should also cite hearing loss and delirium as possible adjunct misdiagnosis of cobaltism (doi: 10.1177/0960327111414280 - doi: 10.3390/jcm12020435). Please briefly discuss and add these 2 references. See lines 61-73 and 124-129

 

Added references

33. Asker S, Asker M, Yeltekin AC, Aslan M, Demir H. Serum levels of trace minerals and heavy metals in severe obstructive sleep apnea patients: correlates and clinical implications. Sleep and Breathing. 2015;19:547-52.

48. Agrup C, Gleeson M, Rudge P. The inner ear and the neurologist. Journal of Neurology, Neurosurgery & Psychiatry. 2007;78(2):114-22.

49. Di Mauro P, La Mantia I, Cocuzza S, Sciancalepore PI, Rasà D, Maniaci A, et al. Acute vertigo after COVID-19 vaccination: case series and literature review. Frontiers in medicine. 2022;8:790931.

26. Asker S, Asker M, Yeltekin AC, Aslan M, Demir H. Serum levels of trace minerals and heavy metals in severe obstructive sleep apnea patients: correlates and clinical implications. Sleep and Breathing. 2015;19:547-52. 40. Agrup C, Gleeson M, Rudge P. The inner ear and the neurologist. Journal of Neurology, Neurosurgery & Psychiatry. 2007;78(2):114-22.

41. Di Mauro P, La Mantia I, Cocuzza S, Sciancalepore PI, Rasà D, Maniaci A, et al. Acute vertigo after COVID-19 vaccination: case series and literature review. Frontiers in medicine. 2022;8:790931.

- Line 58-64. These sentences summarize both methods and results, and should therefore be not included in the introduction section. This content moved to methods.

- Please provide ethical committee approval or waiver code and date of approval. See lines 81-86.

- How did authors retrieve data from patients with dementia? Were they excluded? Please specify. See line 86.

- Please provide the main investigated outcome in the methods section. See lines 90-93.

- Lack of data on cardiovascular function of the patients enrolled should be highlighted as a further possible limitation of the study and direction of future research, as echocardiography is nowadays a feasible bedside tool to investigate systolic and diastolic function of hospitalized patients (doi: 10.2106/JBJS.16.00743- doi: 10.1007/s12630-022-02225-0 - [1]. Please discuss and add these 3 references. See lines 323-326, added references:

52. Berber R, Abdel-Gadir A, Rosmini S, Captur G, Nordin S, Culotta V, et al. Assessing for cardiotoxicity from metal-on-metal hip implants with advanced multimodality imaging techniques. The Journal of bone and joint surgery American volume. 2017;99(21):1827.

53. Sanfilippo F, La Via L, Flower L, Madhivathanan P, Astuto M. The value of subcostal echocardiographic assessment, and directions for future research. Canadian Journal of Anesthesia/Journal canadien d'anesthésie. 2022;69(5):676-7.

---

## [Decision Letter · Decision Letter 1]

16 Nov 2023

The association of Cobalturia with Cobaltism Symptoms

A prospective blinded study of 229 post-arthroplasty patients

PONE-D-23-19885R1

Dear Dr. Tower,

We’re pleased to inform you that your manuscript has been judged scientifically suitable for publication and will be formally accepted for publication once it meets all outstanding technical requirements.

Kind regards,

Antonino Maniaci

Academic Editor

PLOS ONE

Additional Editor Comments (optional):

The paper is improved and can be accepted. Bests

Reviewers' comments:

Reviewer's Responses to Questions

**Comments to the Author**

1. If the authors have adequately addressed your comments raised in a previous round of review and you feel that this manuscript is now acceptable for publication, you may indicate that here to bypass the “Comments to the Author” section, enter your conflict of interest statement in the “Confidential to Editor” section, and submit your "Accept" recommendation.

Reviewer #1: All comments have been addressed

Reviewer #2: (No Response)

2. Is the manuscript technically sound, and do the data support the conclusions?

Reviewer #1: Yes

Reviewer #2: (No Response)

3. Has the statistical analysis been performed appropriately and rigorously? 

Reviewer #1: Yes

Reviewer #2: (No Response)

4. Have the authors made all data underlying the findings in their manuscript fully available?

Reviewer #1: Yes

Reviewer #2: (No Response)

5. Is the manuscript presented in an intelligible fashion and written in standard English?

Reviewer #1: Yes

Reviewer #2: (No Response)

6. Review Comments to the Author

Reviewer #1: All the suggestions required were addressed. I'm glad to hear that the revisions met our expectations. Best regards

Reviewer #2: The manuscript is well written and the authors assessed all the issues presented. I believe the paper is now ready for acceptance.

7. PLOS authors have the option to publish the peer review history of their article (what does this mean?). If published, this will include your full peer review and any attached files.

Reviewer #1: No

Reviewer #2: No

---

## [Editor Report · Acceptance letter]

4 Dec 2023

PONE-D-23-19885R1 

The association of Cobalturia with cobaltism symptoms
A prospective blinded study of 229 post-arthroplasty patients 

Dear Dr. Tower:

I'm pleased to inform you that your manuscript has been deemed suitable for publication in PLOS ONE. Congratulations! Your manuscript is now with our production department. 

Kind regards, 

on behalf of

Prof. Antonino Maniaci 

Academic Editor

PLOS ONE